# The Effects of 1-Deoxynojirimycin from Mulberry on Oxidative Stress and Inflammation in Laying Hens and the Direct Effects on Intestine Epithelium Cells In Vitro

**DOI:** 10.3390/ani13182830

**Published:** 2023-09-06

**Authors:** Mingzhu Wang, Yuan Feng, Tao Li, Chengfeng Zhao, Adileidys Ruiz Barcenas, Boris Ramos Serrano, Liang Qu, Manman Shen, Weiguo Zhao

**Affiliations:** 1College of Biotechnology, Jiangsu University of Science and Technology, Zhenjiang 212100, China; wangmingzhu129@163.com (M.W.); fengyuan6181@126.com (Y.F.); ttaolii@163.com (T.L.); inmitlessness@163.com (C.Z.); 2Plant Protein and Bionatural Products Research Center, Havana 999075, Cuba; adileidys2017@gmail.com (A.R.B.); boris.ramos1972@gmail.com (B.R.S.); 3Jiangsu Institute of Poultry Science, Yangzhou 225212, China; liangquyz@126.com; 4College of Animal Science and Technology, Yangzhou University, Yangzhou 225009, China

**Keywords:** 1-deoxynojirimycin, intestine, chicken, oxidative stress, inflammatory response

## Abstract

**Simple Summary:**

Intestinal health is an integral part of general health. Plants and their extracts, such as mulberry and 1-Deoxynojirimycin, have the potential to improve oxidative stress and inflammatory responses. The effect of 1-deoxynojirimycin (DNJ) extract of mulberry (DNJ-E) and its underlying mechanism regulating intestinal epithelium cells (IEC) cultured in vitro were investigated, analyzing oxidative parameters, inflammatory cytokines, and their related genes. The results revealed that a diet supplemented with 50 mg/kg DNJ-E could influence oxidative stress in layers. In vitro IEC experiments, antioxidative parameters, and IL-1β were significantly changed after 5 µM DNJ treatment. Low levels of DNJ were involved in regulating oxidative stress, leading to cell protection.

**Abstract:**

The intestine is highly vulnerable to various factors and has been proposed as a promising determinant for poultry health. Phytogenic or plant-derived feed additives can be used to help improve intestinal health. In this study, we aimed to investigate the effects of DNJ on the antioxidative parameters, including malondialdehyde (MDA), total superoxide dismutase (T-SOD), catalase (CAT), glutathione peroxidase (GSH-Px), and inflammatory cytokines (IL-6, IL-1β, and TNF-α), in plasma and intestinal tissues using layers supplemented with or without the DNJ extract of mulberry leaves (DNJ-E) via the ELISA method. A total of 192 healthy Hy-Line Brown layers, aged 47 weeks old, were used to conduct a 56-day study. All hens were randomly separated into four groups as follows: a basal diet containing 0 mg/kg DNJ-E(CON), 50 mg/kg, 100 mg/kg, and 150 mg/kg DNJ-E. Furthermore, the potential mechanism by which DNJ influences intestinal function was also investigated in in vitro cultured intestinal epithelium cells (IEC) with quantification methods including the use of a cell counting kit-8 (CCK8), ELISA, qRT-PCR, and ROS detection. The results showed that CAT in plasma significantly increased following 50 mg/kg DNJ-E supplementation. Moreover, 50 mg/kg DNJ-E supplementation was associated with increases in T-SOD in the jejunum and ileum. However, there was no significant difference in inflammatory cytokines between groups in in vivo experiments. Subsequent in vitro IEC studies revealed that cell viability increased significantly following 5 µM and 10 µM DNJ treatments while decreasing significantly following 20 µM DNJ treatment. Antioxidative parameters improved at 5 µM and 10 µM DNJ concentrations. However, there were no ameliorative effects on antioxidant parameters observed under 20 µM DNJ treatment. The expression levels of *Nrf2* mRNA increased significantly under DNJ treatment. DNJ treatment was associated with significant changes in the expression of genes of inflammatory cytokines. In conclusion, our study revealed that DNJ could improve oxidative stress and inflammation responses in the chicken intestine. These findings provide a theoretical reference for the development of functional feed additives that regulate intestinal health and lay the foundation for systematically revealing the mechanism of DNJ.

## 1. Introduction

With the rapid development of genetic selection and intensive management, immense improvements have been made in the breast production of broiler chickens, along with significant achievements in the hen-day egg production in laying hens [1]. However, the modern poultry industry is currently suffering from the virality of biotic or abiotic stress, including shock, extreme temperature, and pathogenic microorganisms [2]. Various challenge factors induce intestinal oxidative stress and increase reactive oxygen species (ROS) accumulation [3,4]. Oxidative stress occurs when the excessive accumulation of ROS exceeds the capacity of antioxidant enzymes such as superoxide dismutase (SOD) and glutathione peroxidase (GSH-Px) [5]. ROS leads to tissue damage and immune dysfunction, destroying the intestinal barrier and leading to flora disorder, and results in impaired growth performance [6]. Provoking intestinal damage with various challenges is an important method of exploring the mechanisms of intestinal damage and finding ways to treat conditions that involve such damage. Hence, boosting the levels of antioxidant enzymes can be convenient for enhancing intestinal function.

For decades, natural plants and plant extracts have been used for intestinal health in poultry [3]. Mulberry (*Morus alba* L.) has been extensively used in the sericulture industry and the feed industry [7,8] due to it being rich in nutrients and biologically active components such as alkaloids, polysaccharides, and flavonoids [9,10]. Notably, 1-Deoxynojirimycin (DNJ), a natural alkaloid, was first discovered in mulberry leaves in 1976 [11], with a content of about 0.1% [12,13]. It has been reported that DNJ significantly affects the treatment of diabetes [14]. With the deepening of research, the biological activity of DNJ has been greatly expanded, including hypoglycemic, antioxidant, anti-inflammatory, antiviral, anti-tumor, and the improvement of gut health [15,16,17]. Animal experiments showed that DNJ from mulberry leaves changed the digestive system of geese and had side effects on body weight [18]. Therefore, the effects of DNJ on intestinal function in chickens should be further explored and expanded.

Since the effects of DNJ on the intestinal health of chickens were not clear, this research was conducted to clarify the effects of the supplementation with the DNJ extract of mulberry leaves (DNJ-E) on the antioxidant capacity and immune response in chickens. Furthermore, intestinal epithelial cells (IEC) cultured in vitro can help in studying the intricate molecular mechanisms by which DNJ affects intestinal function. In this study, we continued to explore the molecular mechanism determinants of DNJ involved in the oxidative stress and inflammatory response of IEC cultured in vitro.

## 2. Materials and Methods

### 2.1. In Vivo Experimental Design, Animals, and Sampling

#### 2.1.1. Animals

The animal experiments were approved by the Institution Animal Care and Use Committee of Jiangsu University of Science and Technology (Zhenjiang, China, No. GSB202131002). A total of 192 Hy-line Brown layers at 47 weeks old with a similar laying rate were accrued from a commercial farm belonging to one of our co-authors (Zhenjiang, Jiangsu, China). The hens were randomly divided into 4 groups (each group had 4 replicates; each replicate had 12 birds) and fed on a corn–soybean diet as the basal diet, and basal diets were supplemented with 50 mg/kg, 100 mg/kg, and 150 mg/kg DNJ extracts from mulberry leaves (DNJ-E, purity 40%), which were bought from Shengqing Biotechnology Co. Ltd., Xi’an, China. The diet composition and nutritional levels are presented in Appendix A and have been shown in our previous study [19]. All the hens were reared in stair-step caging with ad libitum access to water and feed. The light regime was 16 L: 8 D. Detailed information on the management of the hens is provided in our previous study [19].

#### 2.1.2. Sample Collection

The production performance, including the laying rate, feed intake, egg weight, and egg quality, was recorded during the experiment. The feed-to-egg production ratio was calculated. Detailed information on body weight and production performance is presented in Appendix A, which was presented in our previous study by Feng et al. [19]. At the end of the experiment, two hens per replicate were randomly chosen and used for subsequent analysis. First, samples of blood were collected from the wing vein and anticoagulated by heparin sodium to isolate plasma. Plasma was separated via centrifugation at 3000 rpm for 10 min and stored at −20 °C for subsequent assays. Second, after decapitating and dissecting the hens, 3 cm of jejunum and ileum were collected. The samples of the jejunum and ileum were also temporarily stored in liquid nitrogen and then transferred to −80 °C for storage to measure the oxidative stress parameters and inflammatory cytokines.

#### 2.1.3. Antioxidative Stress Parameters and Inflammatory Cytokines Assays

The oxidative stress parameters of superoxide dismutase (T-SOD), catalase (CAT), total glutathione peroxidase (GSH-Px), and malondialdehyde (MDA) in the plasma, jejunum, and ileum were detected using the ELISA method. All samples were homogenized with cold saline at a ratio of 1:10 (*w*/*v*) in a glass homogenizer and counted on a spectrophotometer after being operated according to the manufacturers of the enzyme-linked immunoassay (ELISA) kits from Nanjing Jiancheng Bioengineering Institute (Nanjing, China).

The inflammatory cytokines, including interleukin-1β (IL-1β), interleukin-6 (IL-6), and tumor necrosis factor-alpha (TNF-α), were also measured using the ELISA kits from Nanjing Aoqing Company (Nanjing, China).

#### 2.1.4. Quantitative Reverse-Transcription of PCR (qRT-PCR)

Total RNA was extracted from tissues using Trizol reagent Vazyme Co. (Nanjing, China) according to the manufacturer’s instructions. RNA concentration and purity were determined using a NanoDrop spectrophotometer (Implen, München, Germany). Only RNA samples with a ratio between 1.8 and 2.0 were used for subsequent analysis. The first strand of cDNA was prepared with a commercial kit, the HiScript II Q Select RT SuperMix for qRT-PCR (Vazyme, Nanjing, China), according to the manufacturer’s guidelines. The qRT-PCR reaction was performed using the ChamQ SYBR Color qPCR Master Mix Kit (Vazyme, Nanjing, China) on a CFX96 Real-Time (Bio-Rad, Hercules, CA, USA). The reaction system was as follows: 10 µL 2 × ChamQ SYBR Color qPCR Master Mix, 0.4 µL PCR forward primer (10 µM), 0.4 µL PCR reverse primer (10 µM), 1.0 µL cDNA, and H_2_O to supplement the whole system to 20 µL. The PCR reaction conditions are as follows: 95 °C 30 s, followed by 40 cycles of 95 °C for 10 s and 60 °C for 30 s, then the PCR temperature was increased from 60 to 95 °C to generate the dissolution curve. Genes related to oxidative stress and inflammatory cytokines were determined using *ACTB* as an internal reference gene. The gene amplification efficiency was measured, and only primers with an amplification efficiency of >90% were used. All primer sequences are shown in Table 1. The relative expression levels of mRNA were calculated using the 2^−ΔΔCT^ method.

### 2.2. In Vitro Experimental Design and Sampling

#### 2.2.1. Culture and Identification of Chicken Intestinal Epithelial Cells (IEC)

To further investigate the effects of DNJ on the IEC in vitro, the IEC samples were isolated and cultured from chicken of an embryonic age of 18 (E18) bought from Yangzhou Xianglong Poultry Ltd. (Yangzhou, China). A method for the isolation and culture of IEC has been described in previous studies [20,21] and was improved in this present study. Briefly, the intestine was removed from the embryo and placed in a sterile plate containing D-Hanks (Solarbio, Guangzhou, China) to repeatedly wash it until the washing liquid was clear. With sterilized scissors, intestinal tissue was homogenized and transferred into a centrifuge tube for about 40 min before being digested with collagenase type I (Sangon, Shanghai, China). After complete digestion, the tissues were centrifuged for 10 min at 1000 rpm, and the supernatant was discarded. After resuspending the cells in DMEM/F12 medium (Gibco, Waltham, MA, USA), they were filtered through sieves with a 200-mesh sieve. The unfiltered cells were collected and resuspended in DMEM/F12 medium containing 2% fetal bovine serum (FBS) (Gibco, Waltham, MA, USA). The cell-resuspended solutions were plated and cultured in 37 °C, 5 % CO_2_ incubator.

Cell morphology and specific identification were carried out > 24 h after plating. Alkaline phospholipase staining was performed with an alkaline phosphatase staining kit (Beyotime, Nanjing, China) according to the protocol instruction when the growth density of IEC reached about 80%. Meanwhile, the cells were collected and subjected to qRT-PCR using IEC-specific primers, including E-cadherin (*CHD1*), cytokeratin 18 (*CK18*), and G-protein-coupled receptor 5 (*LGR5*) (Table 1). The chicken embryonic fibroblasts (DF-1) preserved in our lab were used as the negative reference, and the jejunum and ileum from the embryo were used as the positive references.

#### 2.2.2. Measurement of Antioxidative Parameters and Inflammatory Cytokines

Cells were treated with DNJ (Energy-Chemical, Shanghai, China) at the concentrations of 0 (Control), 5 μM, 10 μM, and 20 μM for 24 h. Antioxidative parameters and inflammatory cytokines were tested. Genes related to oxidation stress and inflammation, as shown in Table 1, were also determined using the qRT-PCR method.

#### 2.2.3. Determination of Cell Viability

Cell viability was measured with a cell counting kit-8 (CCK8) assay. Briefly, cells were seeded in a 96-well plate and treated with different concentrations of DNJ. After 24 h, 10 µL CCK8 was added into each well and co-incubated with cells for 2 h at 37 °C. Subsequently, the culture plate absorbance at 450 nm was determined using a microplate reader (Biotek Instruments, Winooski, VT, USA).

#### 2.2.4. Evaluation of Intracellular ROS Content

ROS accumulation in IEC was detected using a ROS detection kit (Beyotime, Nanjing, China). Briefly, the cells were supplemented with diluted DCFH-DA (10 μM) and incubated at 37 °C for 20 min. After incubation, the cells were washed with trypsin I without EDTA (Solarbio, Guangzhou, China) and collected to detect the intracellular ROS content via a microplate reader (Biotek Instruments, Winooski, VT, USA).

### 2.3. Statistical Analyses

At first, the data distribution type was evaluated employing the Kruskal–Wallis test. All data sets showing normal distribution were analyzed using the one-way ANOVA. All values were presented as the mean and standard error of the mean. Data were analyzed using analyses of variance, linear, and quadratic regression with SPSS v.20.0 (IBM SPSS, NY: IBM Corp., Armonk, NY, USA). The LSD test was used to differentiate the difference between means at *p* < 0.05. All data were graphed with GraphPad Prism 8.0.1 software (GraphPad Software, San Diego, CA, USA).

## 3. Results

### 3.1. Effect of DNJ-E on Antioxidant Capacity and Inflammatory Cytokines in Layers

#### 3.1.1. Determination of Antioxidant Capacity Changes upon DNJ-E Supplementation in Layers

Changes in the antioxidant capacity of plasma are summarized in Table 2. Compared with the control group, the CAT activities in plasma increased after DNJ-E supplementation with 50 mg/kg and 100 mg/kg (*p* < 0.05). While the MDA, T-SOD, and GSH-Px showed no statistically significant differences among groups. There is also a statistically significant quadratic effect of DNJ-E on CAT capacity (*p* < 0.05).

The changes in the antioxidant capacity of jejunum are summarized in Table 3. Compared with the control group, the T-SOD activities increased after DNJ-E supplementation with 50 mg/kg and 100 mg/kg (*p* < 0.05), while the MDA, CAT, and GSH-Px showed no significant differences among the groups. The DNJ-E supplementation has a significant quadratic effect on MDA and SOD (*p* < 0.05).

The changes in the antioxidant capacity of the ileum are summarized in Table 4. Compared with the control group, the activities of T-SOD and GSH-Px and the quadratic effect increased after DNJ-E supplementation with 50 mg/kg and 100 mg/kg (*p* < 0.05). The activities of MDA and CAT showed no significant differences among the groups.

#### 3.1.2. Determination of Inflammatory Cytokines upon DNJ-E Supplementation in Layers

Changes in inflammatory cytokines are summarized in Table 5, which showed that there were no significant changes in IL-6, IL-1β, and TNF-α in the plasma, jejunum, and ileum between the untreated group and the treated groups.

### 3.2. Effect of DNJ on Intestinal Epithelial Cell Culture In Vitro

#### 3.2.1. Identification of Intestinal Epithelial Cells from Chicken Embryo

The morphological observation of IEC was performed on the 24 h and 48 h cultures using an optical microscope (Figure 1A,B). With the cultures, time increased the cell morphology, changing the IEC from clumps to the typical ‘paving stone’ shape. As shown in Figure 1C, the blue-purple color represents alkaline phosphatase-stained cells, indicating that the majority of the cells in our cultures are IECs. Additionally, these results were confirmed using the marker genes of *CHD1*, *KRT18*, and *LGR5* via the qRT-PCR method (Figure 1D).

#### 3.2.2. Determination of Cell Viability under DNJ Treatment

As shown in Figure 2, compared with the control group, cell viability was significantly increased under the treatment of 5 μM and 10 μM concentrations of DNJ (*p* < 0.05) and significantly decreased under the treatment of 20 μM concentration of DNJ (*p* < 0.05).

#### 3.2.3. Determination of Intracellular ROS Levels

As shown in Figure 3, compared with the control group, intracellular ROS production was significantly decreased in DNJ treatment groups (at 5 μM, 10 μM, and 20 μM concentrations) (*p* < 0.05).

#### 3.2.4. Detection of Antioxidant Parameters in the Supernatant under the Treatment of DNJ

As shown in Table 6, compared with the control group, the activities of CAT, GSH-Px, and T-SOD increased significantly after 5 µM DNJ treatment (*p* < 0.05), while CAT decreased significantly after 20 µM DNJ treatment (*p* < 0.05). Treatment with 10 µM DNJ significantly increased the activity of CAT and T-SOD and decreased the activity of MDA (*p* < 0.05). Notably, the quadratic DNJ treatment effect was significant in all antioxidant parameters (*p* < 0.05).

There were no significant changes in the marker genes examined except *Nrf2*, which showed a significant increase under DNJ treatment (*p* < 0.05), especially under the 10 µM DNJ treatment group (Figure 4).

#### 3.2.5. Detection of Inflammatory Cytokines under DNJ Treatment

Table 7 showed that the treatment of DNJ with 5 μM and 10 μM concentrations both significantly reduced the content of IL-1β (*p* < 0.05), while IL-6 and TNF-α levels did not significantly change. A significant quadratic effect of DNJ was observed in IL-1β (*p* < 0.05).

The analysis of the marker gene expressions showed that the treatment of DNJ with 10 μM and 20 μM concentrations significantly reduced the mRNA expression levels of *IL-1β* and *IL-6* (*p* < 0.05), but 20 μM DNJ treatment significantly increased the mRNA expression levels of *TNF-α* (*p* < 0.05, Figure 5).

## 4. Discussion

Intestine function changes with nutritional variations, stress, aging, and disease and determines the nutrient absorption capacity. It is also closely related to the antioxidative stress and immune response [22]. Maintaining intestinal health is essential for the health status of poultry. When the body is in an oxidative stress state, there is an increase in MDA and the accumulation of ROS and a decrease in SOD, CAT, and GSH-Px, which aggregates the damage to cells and tissues, causing a series of health problems [23,24]. Excessive oxidative stress often induces inflammation [25] and promotes cells to produce inflammatory cytokines, such as IL-1β, IL-6, and TNF-α, and activate inflammatory responses [26]. Hence, further investigations on antioxidant stress are highly required.

An increasing number of studies showed that various phytogenic feed additives are used to improve poultry intestinal health [27,28,29]. It is well known that DNJ is an effective α-glucosidase inhibitor and has a role in the hypoglycemic effect [30]. The level of DNJ is relatively high in mulberry, except for higher levels of the microorganism [31,32]. With the deepening of the research on DNJ, an increasing number of studies have indicated that DNJ is implicated in the regulation of antioxidants and immunity via multiple signaling pathways [15,16,17,33,34]. Our previous study [19] showed that the organ indexes of the jejunum and ileum weight changed upon the supplementation of DNJ-E. Whether DNJ influences the function of the intestine via oxidative stress and inflammation response needs further investigation. Additionally, the potential regulatory mechanism of DNJ on IEC culture in vitro needs further investigation.

Our data demonstrated that supplementation with the DNJ extract of mulberry in layers affected the antioxidant status of the serum and intestinal organs, as demonstrated by the higher activities of antioxidant enzymes, T-SOD and GSH-Px, under 50 mg/kg DNJ-E supplementation in the basal diet group. However, the high levels of DNJ-E supplementation in the diet did not significantly influence the antioxidant status concerning all antioxidant parameters. Moreover, the treatment of DNJ-E did not significantly affect the inflammatory cytokines. The results were not fully consistent with the previous studies [34,35]. MDA is an important indicator of cellular oxidative damage [36], reflecting the degree of oxidative injury to the organism. The MDA levels in the plasma and intestine in this current study showed no significant difference between the groups, indicating that the hens were not in an oxidative stress condition. All the above characteristics could be the reasons that account for this inconsistency in previous studies.

To further assess the regulatory contribution of DNJ on intestinal epithelial cells (IEC), further evaluation regarding the effect of DNJ on the IEC in vitro is needed. Meanwhile, there has been increasing interest in the in vitro culture of chicken intestinal epithelial cells (IECs), as these cells play critical roles in nutrient absorption, innate immunity, and host–microbe interactions in poultry. However, the establishment of in vitro culture systems for chicken IEC is still challenging [21,37]. In this current study, the IEC culture protocol was set up and optimized based on a study by Ghiselli et al. [20]. Phenotypic characterization via alkaline phosphatase staining and marker genes *CHD1*, *CK18*, and *LGR5* of the identification of IEC indicated that the IECs were successfully isolated and cultured. The successful establishment of in vitro culture systems for chicken IECs will provide a powerful tool for investigating the molecular and cellular mechanisms involved in nutrient absorption, gut development, and mucosal immunity in chickens, as well as for the DNJ role.

In this current study, a pro-proliferation effect was observed for the low levels of DNJ treatment, while an anti-proliferation effect was observed for the high levels of DNJ treatment. It has been proven that DNJ with 10 µM concentration could block the anti-proliferation effect of high-glucose-induced umbilical vein endothelial cells [38]. Previous studies found that the DNJ extract of the mulberry leaves had an adverse effect on the growth performance of geese [18]. The combined results of this current study and the previous literature proved that low levels of DNJ may be advantageous in cell function and body health.

Moreover, in this present study, the treatment of IEC with DNJ in vitro significantly changed the antioxidant parameters and IL-1β and the expression of genes related to oxidative stress and inflammation. Overall, the low levels of DNJ improved antioxidant capacity and anti-inflammatory ability, while the high levels of DNJ repressed them. The antioxidative and anti-inflammatory effects of DNJ were more pronounced in IECs than in vivo experiments in this study. Understandably, the in vivo conditions are more complex than the in vitro conditions. Despite this, we observed that *Nrf2* mRNA significantly increased under DNJ treatment, especially with 5 µM and 10 µM DNJ. Nrf2 has been shown to be the most universal coregulator of Keap1/Nrf2 signaling, which plays a critical role in maintaining cellular homeostasis and protecting cells against oxidative damage [39]. Nrf2 binds to the antioxidant response element (ARE) located in the promoter region of target genes, leading to the regulation of various antioxidant enzymes, such as T-SOD, CAT, and GSH-Px, and the expression of anti-inflammatory cytokines [40], causing a broad and coordinated set of downstream reactions that reduce oxidative stress [41]. From the results of this study, it can be seen that DNJ might have a potential role in the activation of the Keap1/Nrf2 signaling pathway, which, in turn, regulates oxidative stress status and inflammation. Genes related to inflammatory cytokines changing significantly were noted in this study. However, the observation obtained in this current study was under the health state of hens and normal IEC cultured in vitro, which still had some limitations. Further work is needed to analyze the DNJ regulation under oxidative stress conditions induced by heat stress, pathogens, and toxins. If so, the protective effect of DNJ in the oxidative stress condition would be expected to show a better effect.

## 5. Conclusions

Overall, our results indicated that DNJ-E could affect the oxidative stress indicators in the blood, jejunum, and ileum in layers, with the appropriate levels of 50 mg/kg in the basal diet. Inflammatory cytokines were not affected by different treatments with DNJ-E. Furthermore, based on the successful culturing of IEC in vitro, treatment with DNJ significantly reduced the intracellular ROS content and MDA content. The cell viability significantly increased under 5 µM and 10 µM DNJ treatment. Low levels of DNJ improved the activity of the T-SOD, CAT, and GSH-Px antioxidant enzymes and increased the expression levels of *Nrf2* mRNA. DNJ exerts anti-inflammatory effects by reducing the levels of the inflammatory cytokines IL-1β and IL-6 and the expression of genes related to inflammatory cytokines. These results indicate that low levels of DNJ or DNJ-E are beneficial for the intestinal health of chickens.

## Figures and Tables

**Figure 1 animals-13-02830-f001:**
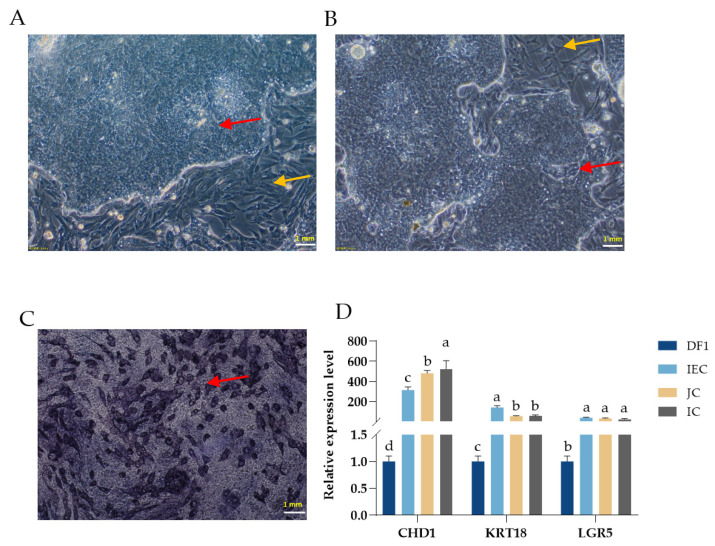
Morphology and identification of intestinal epithelial cell culture in vitro. The images of (**A**,**B**) show the morphology of intestinal epithelial cells (IEC) cultured for 24 h and 48 h observed under a microscope (20×), respectively. (**C**) The IEC staining for alkaline phosphatase staining after culturing IEC 48 h. (**D**) Marker gene expression for *CHD1*, *KRT18*, and *LGR5* was used to identify IEC cells. DF1: fibroblast; JC: jejunum; IC: ileum; Red arrow: IEC; Yellow arrow: DF1. ^a,b,c,d^ means significant difference between groups (*p* < 0.05).

**Figure 2 animals-13-02830-f002:**
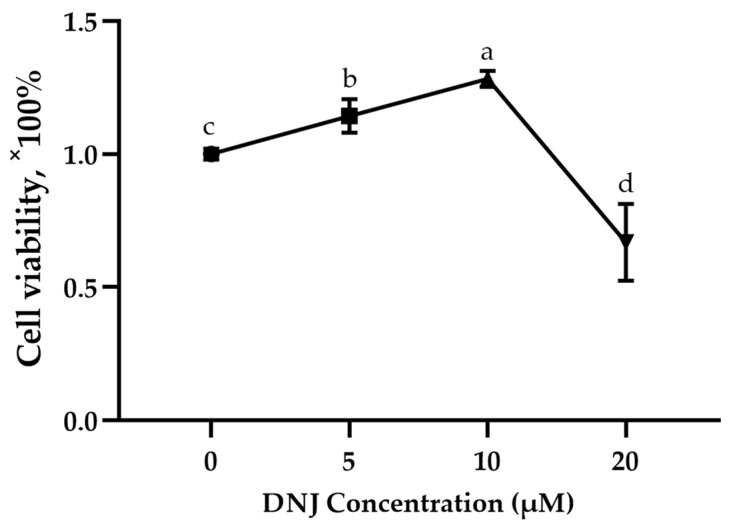
Cell viability of intestinal epithelial cells under treatment of DNJ. ^a,b,c,d^ means significant difference between groups (*p* < 0.05).

**Figure 3 animals-13-02830-f003:**
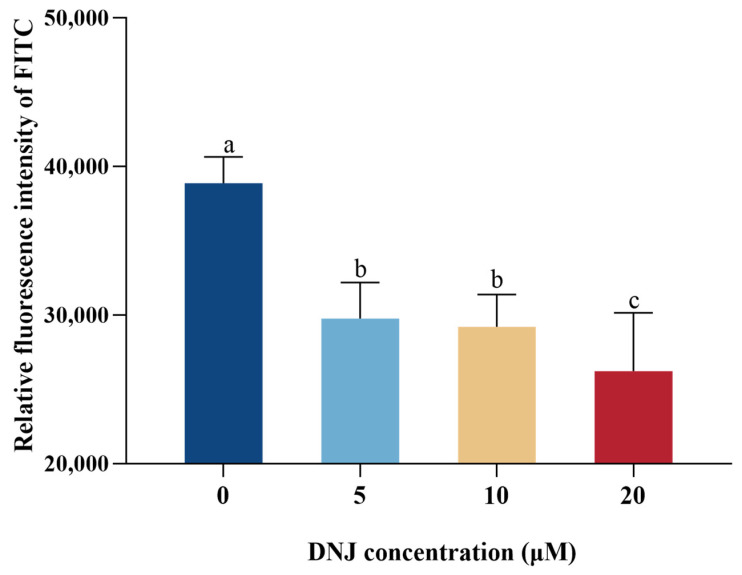
Detection of ROS content in IEC under the treatment of DNJ. ^a,b,c^ means significant difference between groups (*p* < 0.05).

**Figure 4 animals-13-02830-f004:**
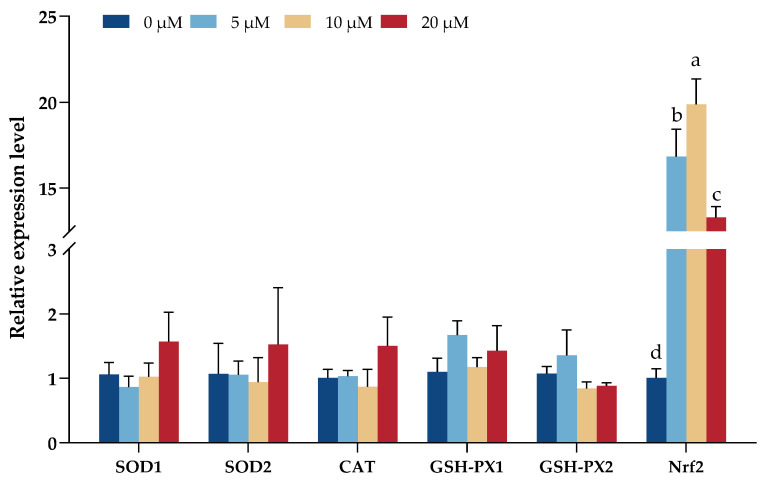
Antioxidant capacity-related gene changes in IEC under the treatment of DNJ. ^a,b,c,d^ means significant difference between groups (*p* < 0.05).

**Figure 5 animals-13-02830-f005:**
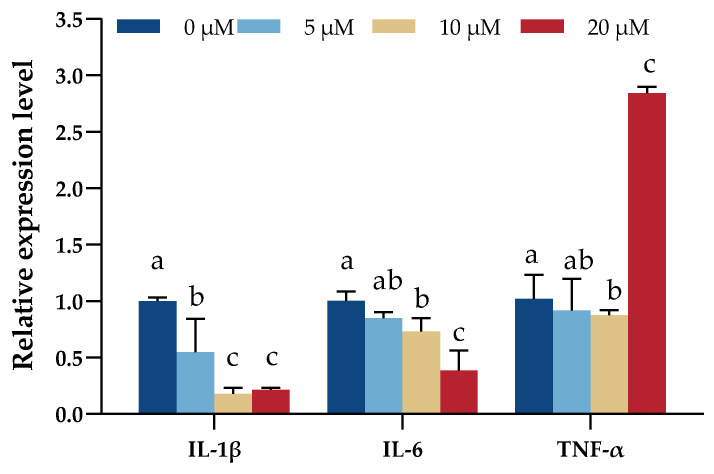
Expression levels of inflammatory cytokines-related genes change in IEC under the treatment of DNJ. ^a,b,c^ means significant difference between groups (*p* < 0.05).

**Table 1 animals-13-02830-t001:** Primer sequences that were used in this study.

Gene Name	Primer Sequence (5′-3′)	Product Length (bp)	Accession No.
*ACTB*	F: CAGCCATCTTTCTTGGGTAT	167	NM_205518.1
	R: CTGTGATCTCCTTCTGCATCC		
*CHD1*	F: TGAAGACAGCCAAGGGCCTG	109	NM_001039258.3
	R: CTGGCGGTGGAGAGTGTGAT		
*KRT18*	F: CACAGATCCGGGAGAGCCTG	110	XM_025145666
	R: CTCCACCGCGCTGTCATAGA		
*LGR5*	F: TGGGCTCCACAGCCTAGAGA	144	XM_425441.5
	R: CCTACAAACGCACGCTCAGG		
*SOD1*	F: TTGTCTGATGGAGATCATGGCTTC	98	NM_205064
	R: TGCTTGCCTTCAGGATTAAAGTGAG		
*SOD2*	F: AGAGGAGAAATACAAAGAGGCG	245	NM_204211.2
	R: AGCCTGATCCTTGAACACCA		
*CAT*	F: TGCAAGGCGAAAGTGTTTGA	158	NM_001031215.2
	R: CCCACAAGATCCCAGTTACCT		
*GSH-Px1*	F: TCACCATGTTCGAGAAGTGC	124	NM_001277853.3
	R: ATGTACTGCGGGTTGGTCAT		
*GSH-Px2*	F: AGGGGGAGAAGGTGGACTT	175	NM_001277854.3
	R: TCCTGGTAGCCGAACTGGT		
*Nrf2*	F: TGACCCAGTCTTCATTTCTGC	186	XM_046921130.1
	R: GGGCTCGTGATTGTGCTTAC		
*IL-1β*	F: CCTCCAGCCAGAAAGTGAGG	109	NM_204524.2
	R: TTGTAGCCCTTGATGCCCAG		
*IL-6*	F: CTCGTCCGGAACAACCTCAA	121	NM_204628.2
	R: AGGTCTGAAAGGCGAACAGG		
*TNF-α*	F: ATCCTCACCCCTACCCTGTC	92	XM_046927265.1
	R: TGTTGGCATAGGCTGTCCTG		

Note: *ACTB*, Actin Beta; *CHD1*, chromodomain helicase DNA binding protein 1; *KRT18*, keratin 18; *LGR5*, leucine-rich repeat-containing G-protein-coupled receptor 5; *SOD1*, superoxide dismutase 1, soluble; *SOD2*, superoxide dismutase 2, soluble; *CAT*, catalase; *GSH-Px1*, glutathione peroxidase 1; *GSH-Px2*, glutathione peroxidase 2; *Nrf2*, Nuclear Factor (Erythroid-Derived 2)-Like; *IL-1β*, interleukin 1, beta; *IL-6*, interleukin 6; *TNF-α*, tumor necrosis factor-alpha.

**Table 2 animals-13-02830-t002:** Effects of DNJ extract from mulberry leaf supplementation on antioxidant capacity of plasma of hens.

Item	Treatment (mg/kg DNJ-E in Basal Diet)				S.E.M.	*p*-Value		
	Control	50	100	150		Treat	Linear	Quadratic
MDA (nmol/mL)	1.58	1.42	1.69	1.50	0.14	0.067	0.312	0.216
SOD (U/mL)	23.35	26.42	25.38	24.33	4.88	0.871	0.731	0.417
CAT (U/mL)	25.32 ^b^	30.12 ^a^	31.23 ^a^	28.42 ^ab^	3.98	0.012	0.071	0.004
GSH-Px (U/mL)	211.14	208.14	206.32	212.45	17.00	0.446	0.600	0.135

^a,b^ means a significant difference between groups (*p* < 0.05). S.E.M. means the standard error of the mean.

**Table 3 animals-13-02830-t003:** Effects of DNJ extract from mulberry leaf supplementation on antioxidant capacity of the jejunum of hens.

Item	Treatment (mg/kg DNJ-E in Basal Diet)				S.E.M	*p*-Value		
	Control	50	100	150		Treat	Linear	Quadratic
MDA (nmol/mL)	1.33	1.42	1.38	1.32	0.05	0.080	0.675	0.020
SOD (U/mL)	8.42 ^b^	10.31 ^a^	9.88 ^a^	7.23 ^bc^	0.61	0.016	0.023	0.001
CAT (U/mgprot)	4.48	5.45	5.21	5.30	0.49	0.086	0.084	0.131
GSH-Px (U/mgprot)	25.32	24.00	25.67	26.32	2.36	0.343	0.572	0.169

^a,b,c^ means a significant difference between groups (*p* < 0.05).

**Table 4 animals-13-02830-t004:** Effect of DNJ extract from mulberry leaf supplementation on antioxidant capacity of the ileum of hens.

Item	Treatment (mg/kg DNJ-E in Basal Diet)				S.E.M	*p*-Value		
	Control	50	100	150		Treat	Linear	Quadratic
MDA (nmol/mL)	0.64	0.57	0.63	0.55	0.06	0.073	0.165	0.254
SOD (U/mL)	26.43 ^c^	29.22 ^a^	28.21 ^b^	28.56 ^b^	2.70	0.034	0.212	0.021
CAT (U/mgprot)	35.22	37.42	34.35	35.44	3.33	0.219	0.332	0.045
GSH-Px (U/mgprot)	50.52 ^c^	55.47 ^a^	53.22 ^b^	50.09 ^c^	3.38	0.009	0.321	0.012

^a,b,c^ means a significant difference between groups (*p* < 0.05).

**Table 5 animals-13-02830-t005:** Effect of DNJ from mulberry leaf extract supplementation on inflammatory cytokines in the serum and intestine of hens.

Organ	Item	Treatment (DNJ-E mg/kg in Basal Diet)				S.E.M	*p*-Value		
		Control	50	100	150		Treat	Linear	Quadratic
Serum	IL-6 (ng/L)	53.22	49.47	51.32	54.33	4.08	0.153	0.231	0.054
	IL-1β (ng/L)	211.37	207.49	217.37	204.22	16.35	0.231	0.187	0.543
	TNF-α (ng/L)	60.11	62.09	58.77	61.31	3.60	0.329	0.124	0.439
Jejunum	IL-6 (ng/L)	5.67	6.54	6.01	5.99	0.66	0.131	0.671	0.129
	IL-1β (ng/L)	88.65	84.32	87.54	89.12	7.15	0.069	0.438	0.039
	TNF-α (ng/L)	3.42	3.01	2.98	3.55	0.44	0.073	0.412	0.046
Ileum	IL-6 (ng/L)	12.11	14.32	11.32	11.10	3.25	0.215	0.327	0.097
	IL-1β (ng/L)	154.32	144.21	148.32	159.11	11.79	0.098	0.341	0.037
	TNF-α (ng/L)	10.55	11.78	11.00	10.00	0.83	0.187	0.268	0.136

**Table 6 animals-13-02830-t006:** Antioxidant parameters change in the supernatant of IEC under the treatment of DNJ.

Item	Treatment (μM DNJ)				S.E.M	*p*-Value		
	Control	5	10	20		Treat	Linear	Quadratic
MDA (nmol/mL)	6.95 ^a^	5.05 ^a^	3.47 ^b^	6.38 ^ab^	0.53	0.047	0.435	0.020
SOD (U/mL)	10.29 ^b^	12.32 ^a^	12.11 ^a^	10.45 ^b^	0.32	0.024	0.897	0.003
CAT (U/mL)	6.31 ^b^	7.93 ^a^	8.3 ^a^	4.83 ^b^	0.44	0.002	0.088	0.001
GSH-Px (U/mL)	76.18 ^b^	78.68 ^a^	77.21 ^ab^	75.88 ^b^	2.31	0.045	0.565	0.051

^a,b^ means significant difference between groups (*p* < 0.05).

**Table 7 animals-13-02830-t007:** Inflammatory cytokines change in IEC under the treatment of DNJ.

Item	Treatment (μM DNJ)				S.E.M	*p*-Value		
	Control	5	10	20		Treat	Linear	Quadratic
IL-6 (ng/L)	126.32	126.66	129.53	118.67	6.91	0.271	0.260	0.170
IL-1β (ng/L)	19.8 ^a^	16.02 ^c^	17.13 ^b^	18.32 ^ab^	1.75	0.016	0.277	0.005
TNF-α (ng/L)	21.55	20.07	21.08	20.30	1.34	0.568	0.471	0.679

^a,b,c^ means significant difference between groups (*p* < 0.05).

## Data Availability

The data presented in this study are available in Appendix A.

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
