# Peer review of "The Effects of 1-Deoxynojirimycin from Mulberry on Oxidative Stress and Inflammation in Laying Hens and the Direct Effects on Intestine Epithelium Cells In Vitro"

_animals, 2023, doi:10.3390/ani13182830_

Round 1
Reviewer 1 Report (Previous Reviewer 2)
This is the corrected version of the previous work showing the biological effects of DNJ extracted from mulberry leaves in relation to oxidative stress and immune response in laying hens.
Unfortunately, some issues still need to be addressed:
Line 101 – which vessel was used to collect the blood?
Line 102 – please explain why duodenum was not included in the study
Line 104 – “samples of jejunum and ileum” instead of “tissues of jejunum and ileum”
Line 163 – both JC and IC are used only one time in the main text. I see no sense to abbreviate it especially that the term jejunum and ileum appeared for the first time in line 103
Line 184 – one-way or two-way ANOVA was used.
Line 184 - the analysis of variance assumes that the data fit the normal distribution. The authors should first check it.
Line 194, 202 etc. – The effect can not be “significant”, can be “statistically significant”
Figure 1 – the images are too small to make any judgement. The figure should be self-explanatory so I suggest authors to add any arrows pointing the observed changes.
Figure 3 – please add unit on Y axis
Tables – please change “um” to “µm”
Author Response
Dear reviewer,
We would like to thank reviewers for the positive and constructive comments and suggestions. We have substantially revised our manuscript after reading the comments provided by you. Our revisions according to editor and the reviewers’ comments are described below. If our revision remains unclear, we kindly ask that you consider making another opportunity on our paper. Thank you so much!
Reply to the comments on Manuscript Number: animals-2531126
Effects of 1-deoxynojirimycin from mulberry on the oxidative stress and inflammation in layers and the direct effects on the intestine epithelium cells in vitro
Reviewer #1: Dear authors,
This is the corrected version of the previous work showing the biological effects of DNJ extracted from mulberry leaves in relation to oxidative stress and immune response in laying hens.
Unfortunately, some issues still need to be addressed:
- Line 101 – which vessel was used to collect the blood?
Response: We used the wing vein for blood collection and added this information in this part.
- Line 102 – please explain why duodenum was not included in the study.
Response: Your suggestion really means a lot to us. At the outset of this experiment, the limitations of our literature mining and experimental design became apparent. The duodenum tissues were not collected and stored at -20 ℃ or -80 ℃ first. But we are still investigating the effects of DNJ on oxidative stress and inflammation in the intestine under conditions of oxidative stress. A better experimental design will be implemented in future research. Thanks again.
- Line 104 – “samples of jejunum and ileum” instead of “tissues of jejunum and ileum”.
Response: Done as requested.
- Line 163 – both JC and IC are used only one time in the main text. I see no sense to abbreviate it especially that the term jejunum and ileum appeared for the first time in line 103
Response: Thanks for you pointing this, we have removed the abbreviate of them.
- Line 184 – one-way or two-way ANOVA was used.
Response: The information was added in Statistical analyses.
- Line 184 - the analysis of variance assumes that the data fit the normal distribution. The authors should first check it.
Response: Sorry for missing this value. All data have been evaluated by distribution type analysis. And the data all showed normal distribution, we have added this information in the Statistical analyses part.
- Line 194, 202 etc. – The effect can not be “significant”, can be “statistically significant”.
Response: Done as requested.
- Figure 1 – the images are too small to make any judgement. The figure should be self-explanatory so I suggest authors to add any arrows pointing the observed changes.
Response: We have added a yellow arrow to represent DF1 and a red arrow to represent IEC in Figure 1. Additionally, we have included the explanatory information in the Figure note.
- Figure 3 – please add unit on Y axis.
Response: Done as requested.
11.Tables – please change “um” to “µm”.
Response: Done as requested.
Thank you for considering our manuscript. All the changes in the revised paper were highlighted in yellow.
Sincerely yours,
Manman Shen, Dr. &Weiguo Zhao Dr.
Reviewer 2 Report (New Reviewer)
Dear authors, this study provides new and important data about the effects of the bacterial metabolite 1-deoxynojirimycin mainly extracted from mulberry leaves on oxidative stress and immune response in layers and its direct effects on the chicken embryo intestine epithelial cells. Moreover, the use of the English language in the present study was appropriate and only some phrases must be rewritten. However, authors must have more publications investigating the effects of phytogenics on layers of antioxidant response and inflammation, and secondly, they must describe better the statistical methods used. Conclusively, this study needs to be minor revised to be published in this journal. The questions which must be answered and the changes which are proposed to be done are presented line by line in the following paragraphs.
Title: The title must be shortened. Also please change, Effect to effects and epithelium cells to epithelial cells.
Simple Summary:
L. 14: Change “intestine health” to intestinal health.
L. 16-18: Unite these two sentences they do not make sense separated.
Abstract:
L. 25: Change “intestine” to intestinal and effect to effects again.
Introduction:
--------------------------
Materials and Methods:
L. 183-186: Why authors did not compare means using Tukey's honestly significant difference (HSD) multiple comparison procedure which is more accurate? And secondly, did the authors conduct no parametric tests?
Results:
----------------------------------------------------------
Discussion:
L. 286-287: Authors must add more publications investigating the effects of phytogenics on layers of antioxidant response at the molecular level. For example, recent research is:
Evangelos C. Anagnostopoulos, Ioannis P. Brouklogiannis, Eirini Griela , Vasileios V. Paraskeuas and Konstantinos C. Mountzouris. 2023. Phytogenic Effects on Layer Production Performance and Cytoprotective Response in the Duodenum. Animals 2023, 13, 294. https://doi.org/10.3390/ani13020294.
I believe that authors can find more.
Conclusions:
------------------------------------------------
Author Response
Dear reviewer,
We would like to thank you for your positive and constructive comments and suggestions. We have made significant revisions to our manuscript based on the comments you provided. The revisions made based on your comments are described below. If our revision is still unclear, we kindly ask that you consider giving us another opportunity to work on our paper. Thank you so much!
Reply to the comments on Manuscript Number: animals-2531126
Effects of 1-deoxynojirimycin from mulberry on the oxidative stress and inflammation in layers and the direct effects on the intestine epithelium cells in vitro
Reviewer #2: Dear authors, this study provides new and important data about the effects of the bacterial metabolite 1-deoxynojirimycin mainly extracted from mulberry leaves on oxidative stress and immune response in layers and its direct effects on the chicken embryo intestine epithelial cells. Moreover, the use of the English language in the present study was appropriate and only some phrases must be rewritten. However, authors must have more publications investigating the effects of phytogenics on layers of antioxidant response and inflammation, and secondly, they must describe better the statistical methods used. Conclusively, this study needs to be minor revised to be published in this journal. The questions which must be answered and the changes which are proposed to be done are presented line by line in the following paragraphs.
Response: Thank you for your insightful comments and suggestions on our manuscript. Therefore, we have modified the text as required. The manuscript has been polished by English pre-edit services from MDPI, see the certification of “English-Editing-Certificate-70108.pdf”. See below for the other details.
Title: The title must be shortened. Also please change, Effect to effects and epithelium cells to
epithelial cells.
Response: Thanks for your suggestions. We have modified title as “Effects of 1-deoxynojirimycin from mulberry on the oxidative stress and inflammation in layers and the direct effects on the intestine epithelium cells in vitro”. If this is not right, could you please give us another chance, thank you very much.
Simple Summary:
- 14: Change “intestine health” to intestinal health.
Response: Done as requested.
- 16-18: Unite these two sentences they do not make sense separated.
Response: Done as requested.
Abstract:
- 25: Change “intestine” to intestinal and effect to effects again.
Response: Done as requested.
Materials and Methods:
- 183-186: Why authors did not compare means using Tukey's honestly significant difference (HSD) multiple comparison procedure which is more accurate? And secondly, did the authors conduct no parametric tests?
Response: We acknowledge and appreciate your comments and suggestions, which are valuable in improving the quality of our manuscript. The ANOVA analysis was followed by a Tukey test when there was the same number of samples in each group or treatment, but some missing values in individual groups. Under this condition, LSD analysis would be better. We reanalyzed our data from groups with the same samples and found that the results obtained using the LSD and Tukey methods were largely consistent. Some results about CAT in serum are listed below.
One-way ANOVA
LSD method
Tukey method
Discussion:
- 286-287: Authors must add more publications investigating the effects of phytogenics on layers of antioxidant response at the molecular level. For example, recent research is:
Evangelos C. Anagnostopoulos, Ioannis P. Brouklogiannis, Eirini Griela , Vasileios V. Paraskeuas and Konstantinos C. Mountzouris. 2023. Phytogenic Effects on Layer Production Performance and Cytoprotective Response in the Duodenum. Animals 2023, 13, 294. https://doi.org/10.3390/ani13020294.
Response: Thanks for your kind suggestion. After reading the paper you suggested, we now recognize the significant effects of phytogenics on layers, as cited in this section.
Thank you for considering our manuscript. All the changes in the revised paper were highlighted in yellow.
Sincerely yours,
Manman Shen, Dr. &Weiguo Zhao Dr.

Reviewer 3 Report (New Reviewer)
Dear authors,
First of all, I would like to congratulate all the authors for their extraordinary work in this section of “Animal Nutrition”. You have included all the necessary information in this article.
The manuscript was well-written and the content was informative and well-presented. I commend the authors for the comprehensive and systematic review of the topic. The manuscript will be a valuable contribution to this journal but after some extensive English editing and some minor revisions.
However, I’ve mentioned some minor corrections which need to be corrected in the comment section of the main manuscript file. Some of these include here:
Line 3-4: Please revise the title of this manuscript.
Line 14-15: Please revise these lines.
Line 43-44: Please add one line at the end of the abstract, which basically explains the basic output of this study and the future recommendations related to this study work as well.
Line 60-61: Please provide some references to support this statement.
Line 102: centrifugation at which rpm for how many minuets to seperate this plasma?
Line 104: Please indicate the tissue sample size here as well.
Line 274-275: Please indicate the software name in the footnote, which you have used for these graphs??
Line 277-278: Please provide some references here to support this statement?
Most importantly, Please check the font style, format, and color throughout the manuscript and please try to set it according to the journal's formatting guidelines.
Best wishes

Dear Authors,
I have thoroughly gone through the manuscript. The manuscript is well based on the novel research on mulberry leaf extract. I have seen many anomalies regarding the English language, especially, since there is a big issue of spaces between the words in many places in the whole of the manuscript. The sentences were too long which may divert the intentions of the readers in the manuscript. Please revise carefully.
Author Response
Dear reviewer,
We would like to thank you for the positive and constructive comments and suggestions. We have substantially revised our manuscript after reading the comments provided by you. Our revisions according to your comments are described below. If our revision remains unclear, we kindly ask that you consider making another opportunity on our paper. Thank you so much!
Reply to the comments on Manuscript Number: animals-2531126
Effects of 1-deoxynojirimycin from mulberry on the oxidative stress and inflammation in layers and the direct effects on the intestine epithelium cells in vitro
Reviewer #3: Dear authors,
First of all, I would like to congratulate all the authors for their extraordinary work in this section of “Animal Nutrition”. You have included all the necessary information in this article.
The manuscript was well-written and the content was informative and well-presented. I commend the authors for the comprehensive and systematic review of the topic. The manuscript will be a valuable contribution to this journal but after some extensive English editing and some minor revisions.
Response: Thanks for your kind comments. We have polished our manuscript by employing MDPI service, see the file of “English-Editing-Certificate-70108.pdf”.
However, I’ve mentioned some minor corrections which need to be corrected in the comment section of the main manuscript file. Some of these include here:
Response: We appreciate the comments and have revised as suggested.
1.Line 3-4: Please revise the title of this manuscript.
Response: Thanks for your kind suggestion. We have revised the title as “Effects of 1-deoxynojirimycin from mulberry on the oxidative stress and inflammation in layers and the direct effects on the intestine epithelium cells in vitro”. If this is not correct, could you please give us another opportunity? Thank you very much.
- Line 43-44: Please add one line at the end of the abstract, which basically explains the basic
output of this study and the future recommendations related to this study work as well.
Response: Thanks for your suggestions, we have added this information at the end of abstract.
- Line 43-44: Please provide some references to support this statement.
Response: We apologize for not understanding this comment. L43-44 is in the abstract, which doesn't require a reference. Which lines are you referring to? Please don't hesitate to point out any mistakes for us.
- centrifugation at which rpm for how many minuets to separate this plasma?
Response: Sorry for our mistake. We modified this issue; the centrifugation rpm was 3000 for 10 min.
5.Line 104: Please indicate the tissue sample size here as well.
Response: Sorry for our mistake. The sample size was 8, we have added this information.
6.Line 274-275: Please indicate the software name in the footnote, which you have used for these
graphs?
Response: All data were graphed by GraphPad Prism. We have added this information in the methods part.
7.Line 277-278: Please provide some references here to support this statement?
Response: Thanks for your kind suggestions. We have provided some related references to support this statement.
Thank you for considering our manuscript. All the changes in the revised paper were highlighted in yellow.
Sincerely yours,
Manman Shen, Dr. &Weiguo Zhao Dr.
This manuscript is a resubmission of an earlier submission. The following is a list of the peer review reports and author responses from that submission.
Round 1
Reviewer 1 Report
1. Line 100. add the animal numbers per replicate.
2. discussion should be related to the performance and intestines growth with the results from reference 19.
Reviewer 2 Report
The manuscript is nicely done and written. The study design is appropriate and apparently, the analyses were carefully performed. I believe that the results are valuable for the scientific community and has significant scientific merit, as it will probably ignite many further studies in the near future.
However, some points need to be clarified before the publication.
Line 24 – Please explain what MDA, T-SOD, CAT, and GSH-Px stand for?
Line 50 – the intestine is an organ not a place.
Line 74 – please present your hypothesis correctly.
Line 102 – what is “The intestine of jejunum and ileum”? Both jejunum and ileum are parts of the small intestine. Another question is why the duodenum was not included in the study?
Line 131 – Did the authors determine primers amplification efficiency?
Line 145 – please follow Journal citation style.
Figure 5 - scale bars are missing.
Figure 5D - In figure 5D, DC stands for duodenum!!! According to MM this part of small intestine was not studied!